# Artificial Intelligence in Digestive Endoscopy—Where Are We and Where Are We Going?

**DOI:** 10.3390/diagnostics12040927

**Published:** 2022-04-08

**Authors:** Radu-Alexandru Vulpoi, Mihaela Luca, Adrian Ciobanu, Andrei Olteanu, Oana-Bogdana Barboi, Vasile Liviu Drug

**Affiliations:** 1Institute of Gastroenterology and Hepatology, Saint Spiridon Hospital, “Grigore T. Popa” University of Medicine and Pharmacy, 700111 Iași, Romania; vulpoi.radu@yahoo.ro (R.-A.V.); olteanuandrei@yahoo.com (A.O.); vasidrug@email.com (V.L.D.); 2Institute of Computer Science, Romanian Academy—Iași Branch, 700481 Iași, Romania; mihaela.luca58@iit.academiaromana-is.ro (M.L.); adrian.ciobanu@iit.academiaromana-is.ro (A.C.)

**Keywords:** artificial intelligence, digestive endoscopy, computer-aided diagnosis, computer-aided detection, deep learning

## Abstract

Artificial intelligence, a computer-based concept that tries to mimic human thinking, is slowly becoming part of the endoscopy lab. It has developed considerably since the first attempt at developing an automated medical diagnostic tool, today being adopted in almost all medical fields, digestive endoscopy included. The detection rate of preneoplastic lesions (i.e., polyps) during colonoscopy may be increased with artificial intelligence assistance. It has also proven useful in detecting signs of ulcerative colitis activity. In upper digestive endoscopy, deep learning models may prove to be useful in the diagnosis and management of upper digestive tract diseases, such as gastroesophageal reflux disease, Barrett’s esophagus, and gastric cancer. As is the case with all new medical devices, there are challenges in the implementation in daily medical practice. The regulatory, economic, organizational culture, and language barriers between humans and machines are a few of them. Even so, many devices have been approved for use by their respective regulators. Future studies are currently striving to develop deep learning models that can replicate a growing amount of human brain activity. In conclusion, artificial intelligence may become an indispensable tool in digestive endoscopy.

## 1. Introduction

Artificial intelligence (AI) is a computer model created to mimic human behavior [1]. In various medical fields, this technology has made its presence felt, in many cases improving diagnosis, treatment, and disease follow-up procedures.

The first attempt to automate medical diagnosis and treatment recommendations was the well-known MYCIN study [2,3]. It was a backward chaining expert system that used AI to identify bacteria and recommend antibiotics (hence the name, MYCIN). 

In the early years of AI, many classic methods were employed, from rule-based systems to neural networks, statistical methods, signal and image processing, some of them using fuzzy, probability, possibility, or chaos theories. These were mainly off-line due to time-consuming computing. The big step was made in the last decade when the diversification of machine learning and deep learning structures was sustained by the development of new devices using parallel computing and multi-core graphics processing units (GPUs) [4].

Today we use real-time tools to assist various medical procedures that researchers have struggled to develop for over half a century.

Devices such as the NVIDIA Jetson microsystems series, NVIDIA GPU boards, etc., are making a difference in the field of real-time medical applications.

## 2. Artificial Intelligence in Colonoscopy

GLOBOCAN, the online database that provides statistical information on 36 types of cancer in 185 countries, was updated in 2020 by the International Agency for Research on Cancer (IARC) [5]. The research estimated over 19 million new cases, all cancers, both sexes, and all ages. Breast cancer was the most commonly diagnosed cancer (11.7% of all cases), followed by lung (11.4%), colorectal (CRC, 10%), prostate (7.3%), stomach (5.6%), liver (4.7%), cervix uteri, (3.1%), esophagus (3.1%) cancers and other cancers for the remaining 42.9% [6]. The mortality data showed that lung cancer was still the leading cause of cancer death (18%), followed by CRC (9.4%), liver (8.3%), stomach (7.7%), breast (6.9%), esophagus (5.5%), pancreas (4.7%) and prostate (3.8%) cancers. Almost 10 million cancer deaths occurred in 2020 [6].

CRC is a major public health problem. According to GLOBOCAN 2020 estimates, CRC is still the third most commonly diagnosed cancer and the second leading cause of cancer death, being responsible for over 900,000 deaths worldwide in 2020 [6,7]. More concerning is the fact that recent studies found that the pattern of CRC incidence is changing, noting a rise in early-onset cases, especially in high-income countries. Although a direct cause has not been found, the most likely culprit would be early-age exposure to large bowel carcinogens [8].

The diagnosis and management of many diseases have improved because of the technological advancements in the healthcare industry and easier access to large medical databases for research. Integrating new technologies into clinical practice may be a key factor. CRC diagnosis still relies on colonoscopy despite significant advances in the field. This investigation is considered essential in reducing CRC incidence and mortality [9]. Early detection is becoming increasingly important for both the medical community and the public. Many CRC screening programs have been launched since 2007. Fifteen of 28 European countries underwent population-based CRC screenings in 2019 [10].

A screening program is considered successful when an early CRC diagnosis is made, and precancerous lesions are diagnosed and treated. At the same time, such a program may be subject to multiple limitations. The increased number of procedures per endoscopist might be correlated with the increased adenoma miss rate, especially for diminutive polyps (≤5 mm). Moreover, during a busy work schedule, operator fatigue is associated with poorer colonoscopy performance. This may negatively affect the polyp detection rate, resulting in a low adenoma detection rate (ADR) [11]. In order to objectively assess the colonoscopy performance, criteria such as preprocedural colon preparation rate, cecal intubation rate, correct identification and management of lesions, and correct postprocedural follow-ups have to be considered [12]. It appears that technological advancement, including AI, might be a solution to these limitations.

The use of AI for colonoscopy has been studied over the last decade. Numerous papers highlight the benefits of implementing such an application in current medical practice [13]. According to current European Society of Gastrointestinal Endoscopy (ESGE) guidelines, all polyps larger than 5 mm need to be removed and sent for histopathology analysis to maximize screening effectiveness. The same thing should be applied to all diminutive polyps (<5 mm) with adenomatous structures. Diminutive polyps located in the rectum and the sigmoid characterized as hyperplastic by a high confidence optical diagnosis tool can be “left in situ” or “resect and discard” [14,15]. Most digestive endoscopists resect polyps and send them to histopathology because optical diagnosis tools are only available in expert medical institutions. Costs may increase, and physicians may become fatigued as a result. Technology development, including AI integration, may reduce this setback in the long run. 

In endoscopy, AI has introduced two concepts: computer-assisted detection (CADe) and computer-assisted diagnosis (CADx). With CADe, the AI model acts as a “second pair of eyes” for the colonoscopist [16]. The capacity of identifying polyps may help to increase the investigator’s ADR, especially for diminutive polyps. This technology is especially useful for beginner colonoscopists, which may achieve results similar to experts in colonoscopy [17,18]. This eliminates the need for immediate endoscopic reassessment and therefore reduces healthcare costs [19]. A meta-analysis published in 2021 highlighted the advantages of using CADe compared to other colonoscopy methods such as high-definition white-light endoscopy, chromoendoscopy (dye-based or Narrow Band Imaging [NBI]), or add-on devices (i.e., systems that increase mucosal visualization, such as full-spectrum endoscopy (FUSE or G-EYE balloon endoscopy). This study showed that CADe is superior to high-definition white-light colonoscopy (an increase in ADR by 7.5%). Furthermore, both chromoendoscopy and increased mucosal visualization systems achieved better ADR (4.4% and 4.1%, respectively) compared to high-definition white-light colonoscopy. Cross-comparison of CADe with chromoendoscopy and increased mucosal visualization systems showed higher ADR with CADe (OR 1.45 [95% CI 1.14–1.85]; moderate certainty of the evidence, and OR 1.54 [95% CI 1.22–1.94]; low certainty of the evidence, respectively) [20].

A CADx system is used to classify a lesion based on several morphological parameters (surface, vascular patterns, shape, size, location), thus generating probability scores for malignancy or non-malignancy [21]. A paper published in 2021 [22] showed that CADx using white light colonoscopy has a sensitivity of 95.5% and a specificity of 84.4%, resulting in an accuracy of 93.2%. CADx using blue light colonoscopy showed slightly superior results: sensitivity 96.3%, specificity 88.7%, and accuracy 94.7%. These data highlight the ability of CADx systems to diagnose the polyp type and thus help with the correct management. Moreover, it may enable one to decide whether to resect a lesion and send it to pathology, resect and discard, or even leave it in place and monitor it over time [14]. More studies are necessary before such an approach may be recommended.

NBI (Narrow Band Imaging)-CADx attracted the most attention from the research community. Numerous studies highlighted the potential benefits of these two technologies working together. In 2018 Chen et al. developed an NBI—AI system capable of differentiating polyps with a sensitivity of 96.3%, specificity of 78.1%, and accuracy of 91% [23]. Zachariah et al. published a paper in 2020 where they wrote about the development of their NBI-CADx system based on convolutional neural networks. The AI model managed to exceed the ASGE PIVI (American Society of Gastrointestinal Endoscopy Preservation and Incorporation of Valuable endoscopic Innovations) standard for both “resect and discard” and “diagnose and leave” strategies. This AI system achieved an accuracy of 94% with NBI, being able to correctly classify diminutive polyps, irrelevant of the endoscopists’ experience [24]. In 2020 Song et al. developed a CADx model for predicting colorectal polyp histology on NBI pictures. In the study, they included both trainees and NBI expert endoscopists. The AI system proved to have higher diagnostic accuracy than the trainees (81–82% vs. 63.8–71.8%) and similar results compared to expert endoscopists. This study concluded that the CADx-NBI system could be an important tool for improving the trainees’ diagnostic accuracy [25]. Similarly, Jin et al. showed that their AI system for NBI achieves high accuracy in distinguishing adenomas from hyperplastic polyps. Moreover, the trainees achieved a near-expert level of accuracy without needing to undergo extensive training [26].

Chromoendoscopy has not been so appealing for AI development. In this case, the pit pattern classification is dependent on the depth of color. This means the more dye is sprayed, the better the quality. Because it is difficult to obtain a uniform image quality, the CADx system may have a hard time learning these patterns [27]. However, an old study led by Takemura et al. in 2010 achieved an accuracy of 98.5% with their CADx model associated with chromoendoscopy [28].

Endocytoscopy is a novel in vivo imaging technique capable of offering a microscopic-like view of the mucosa in real time. Due to the large quantity of information extracted by this technique (i.e., cellular and microvascular patterns), the integration of an AI system in endocytoscopy seems to be an ideal connection. Furthermore, because endocytoscopy provides mostly focused, fixed-size images, the CADx system has an easier job analyzing the images [27]. Misawa et al. developed an NBI-CADx for endocytoscopy with a sensitivity, specificity, and accuracy of 84.5%, 97.6%, and 90.0%, respectively. Moreover, they achieved a probability of diagnosis greater than 90%, which is the ASGE PIVI threshold for a “high-confidence” diagnosis [29]. In 2020 Mori et al. published an article on the cost savings in colonoscopy with CADx systems. They used their AI system with endocytoscopy on 207 patients with 240 diminutive rectosigmoid polyps. The AI correctly differentiated neoplastic polyps with a sensitivity of 93.3%, specificity of 95.2%, and a negative predictive value (NPV) of 95.2%. Having an NPV > 90%, which is the ASGE threshold, the research team was confident in applying the strategy of diagnose-and-leave to all non-neoplastic lesions. As a result, 105 polyps were removed, and 145 polyps were left in place. The study estimated a reduction of the average colonoscopy cost and the annual reimbursement for colonoscopy by 18.9% (US$ 149.2 million) in Japan and 10.9% (US$ 85.2 million) in the United States, compared with the resect-all-polyps strategy [30]. Although the potential cost reduction is impressive, endocytoscopy is not widely used in clinical practice.

A colorectal cancer screening program is effective if the colonoscopy quality indicators are met. Currently, these indicators are subject to intentional or unintentional manipulation. They include pre-procedural indicators (i.e., the rate of adequate bowel preparation, the time interval for colonoscopy, the indication for colonoscopy), procedure indicators (i.e., cecal intubation rate, ADR, withdrawal time, polyp detection rate, management of pathology and complications, patient experience) and post-procedural indicators (appropriate post-polypectomy surveillance) [12]. Using AI together with other computerized management systems, these parameters may be objectively assessed and may provide a procedure quality score. This might be the next step towards achieving higher healthcare performance.

Inflammatory bowel disease (IBD) has also been a focus of AI development in colonoscopy [31]. Several studies were aimed to assess the value of AI during colonoscopy in ulcerative colitis (UC) due to its convenient location in the colon. Studies by Ozaw et al. [32] and Stidham et al. [33] highlighted the benefits of AI in distinguishing active disease from remission during standard colonoscopy. Another study conducted by Bossuyt et al. [34] used a red density score which correlated mucosal redness with the severity of the inflammation. The results were biopsy-confirmed and were consistent with Robart’s histological index [35], the Mayo endoscopic score, and the Ulcerative Colitis Endoscopic Index of Severity (UCEIS) [36].

In Crohn’s Disease (CD), given the various locations of the lesions, video capsule endoscopy (VCE) has been the target for research when trying to implement an AI system [31]. Klang et al. [37] and Barash et al. [38] developed a CAD system capable of detecting ulcerations during VCE and concluded that AI might prove beneficial in CD diagnosis and monitoring. Another interesting study conducted by Ding et al. [39] concluded that the use of AI reduces the reading time of VCE from an average of 96.6 min to 5.9 min.

## 3. Artificial Intelligence in Upper Digestive Endoscopy

The majority of AI research appears to be focused on improving lower digestive endoscopy examinations, but there is a growing interest in upper digestive endoscopy (UDE) as well. Applying AI to diseases such as gastroesophageal reflux disease (GERD), Barrett’s esophagus (BE), and gastric cancer may have great healthcare benefits if newer technologies are implemented.

GERD is a fairly common disease with symptoms ranging from occasional chest discomfort to severe heartburn and regurgitation. In 2021 Wang, C.-C. et al. [40] proposed a deep learning model, named GERD-VGGNet, that managed to identify and classify GERD according to Los Angeles classification both in conventional and in NBI endoscopy. According to the study, their AI model resulted in an accuracy of 87.9%, which was significantly higher than the results of the trainees (75.0% and 65.6%). These results prove that AI may be beneficial for the recognition of GERD disease. However, more research in this field is needed [40].

Barrett’s esophagus is usually the outcome of long-lasting untreated GERD. Because it is a pre-malignant lesion, research has been conducted, and guidelines have been developed for its detection, diagnosis, and follow-up [41]. Numerous recent studies demonstrated that AI might be the missing link for the optimal management of this disease. In 2020, Hashimoto R et al. [42] developed a convolutional neural network (CNN) algorithm designed to differentiate between dysplastic and non-dysplastic lesions in BE. It detected early neoplasia with 95.4% accuracy [42]. Further, in 2019, de Groof et al. [43] developed a CADe system capable of identifying neoplasia in patients with BE. Data have demonstrated that their system yielded better results than those of non-expert endoscopists [43]. Thus, AI may prove to be extremely beneficial to beginner endoscopists. Detecting early neoplasia in BE is a top priority, and AI research may add significant benefits in the future.

Similar to colorectal cancer, gastric cancer is still a major healthcare problem with a poor prognosis if left undiagnosed and untreated [44]. Upper digestive endoscopy (UDE) is the most important procedure to evaluate and diagnose gastric cancer and take biopsy samples. During UDE, novice endoscopists may not be able to correctly assess the entire mucosa. The unseen areas are called blind spots and may hide potential neoplastic lesions. A capable AI system may overcome this impediment and possibly enhance the quality of UDE. In a 2021 study, Wu L et al. [45] developed an AI model named ENDOANGEL. They showed that the number of blind spots per endoscopist was significantly reduced when using the AI model. This technology could detect gastric neoplasia with an accuracy of 84.7% [45].

## 4. Challenges in Implementing AI Systems in Healthcare

Regulations regarding safety, efficacy, and transparency must be approved before AI technology can be used in clinical settings [46]. It is equally important to consider the potential negative patient outcomes. These steps, although they may take time, are absolutely necessary for implementation [46].

AI systems require ongoing maintenance, large data incorporation, software updates, and hardware repairs. All of these activities require human resources and funding support. The economic burden may be significant. To decide if purchasing an AI system is advantageous for patient care, it is necessary to carefully balance investment and benefits. Data regarding AI technology implementation costs are scarce at present [47].

Integrating AI technology into existing systems, such as electronic health record databases, is an important issue. Adapting an AI model to a variety of clinical situations and gaining benefits at an organizational level can be challenging. Creating an AI model that is compatible with a large number of healthcare systems while still being relevant on an individual basis is not an easy task [48]. 

A further challenge is creating a good communication channel between the AI model and physicians. Essentially, this means translating digital information into the usual medical language in order to aid in diagnosis and treatment. The 2021 study by Fonollà R et al. [49] highlighted the possibility of creating an AI system that can overcome this difficulty. They used a large database that included two units: one unit housed polyp images and characteristics according to multiple classifications such as PARIS, NICE, KUDO, and BASIC. The second unit contained medical statements of experts in endoscopy that described the polyps. Common ground was established between the experts regarding the range of accepted terminology when describing polyps. Fonollà R et al. managed to create an AI system that automatically generates a textual polyp description based on the BASIC classification [49]. A step forward has been made, but more research is needed for improving collaboration between AI systems and physicians.

It is important to stress that in order to create an AI medical system, the most important role is played by the medical experts providing the necessary knowledge in a specific field. For instance, in the case of polyp detection during colonoscopy, it is necessary to gather a large variety of saved video colonoscopies performed by top endoscopists. To incorporate knowledge, medical experts must select colonoscopy frames that best represent different types of polyps with different sizes and in different circumstances. Then they have to assist in the important task of annotating colonoscopy frames so that the AI system is able to correctly learn how to detect polyps. The quality of the databases behind an AI system is of paramount importance for the effectiveness of its operation and is given by the value of the medical team that assisted in its development.

## 5. AI Systems Currently Approved for Use in Gastroenterology

Despite numerous difficulties, many AI systems have been validated and approved for use in medical practice. Healthcare corporations, together with their academic partners, were able to develop, test, and evaluate the results of incorporating such devices in clinical practice. Some of the currently available are Endo AID, Olympus Corp., Tokyo, Japan; CAD EYE, Fujifilm Corp., Tokyo, Japan; Discovery, Pentax Corp., Tokyo, Japan; GI Genius, Medtronic Corp., Dublin, Ireland; EndoBRAIN, Cybernet Corp., Tokyo, Japan (Table 1) [50]. The European Medicines Agency (EMA) and the Japanese Pharmaceuticals and Medical Devices Agency (PMDA) approved the use of these AI systems during 2018–2020. A great deal of interest has been expressed in developing and implementing these devices, as shown by the rapid approvals from regulatory agencies [51].

Although the United States FDA was a little more rigid when evaluating AI technology in endoscopy compared with the EMA and the PMDA, on 12 April 2021, they approved the first computer-aided polyp detection system represented by GI Genius, Medtronic Corp [52].

In a multicenter randomized trial in 2020, Repici et al. used GI Genius from Medtronic Corp. The aim of their study was to compare CADe colonoscopy to conventional colonoscopy. Their primary outcome was ADR, and their secondary outcomes were adenoma detection per colonoscopy, non-neoplastic resection rate, and withdrawal time. Using GI Genius, they concluded that the ADR (54.8% vs. 40.4%) and adenomas per colonoscopy were significantly higher in the CADe group compared to the control group. No significant difference was reported regarding withdrawal time or proportion of subjects with resection of non-neoplastic lesions [16].

Olympus’s Endo-AID was presented to the public for the first time during the October 2020 United European Gastroenterology (UEG) Week [53]. The system promises excellent improvement in polyp detection and the potential to reduce the strain on the endoscopist by reducing the need for excessive eye movement. Chieh Sian Koo et al. published in Endoscopy 2021 an article that demonstrated the performance for both CADe and CADx of this AI system [54]. Even so, system improvements are still needed, as shown by a case report published in Endoscopy 2022 by Lafeuille Pierre et al. In this report, the AI system failed to correctly detect a 2.5 cm pseudo-depressed non-granular laterally spreading tumor of the transverse colon, which the pathological examination suggested being an adenocarcinoma [55].

Fujifilm’s CAD EYE was officially presented during the ESGE Connect 2020. In 2021, Helmut Neumann et al. [56] used CAD EYE in combination with a linked color imaging (LCI) technique on 240 polyps that covered the entire spectrum of adenomatous histology. In their study, the AI system achieved a sensitivity of 100% without missing a single lesion. They calculated a 0.001% false-positive frame rate. Above all, the AI system managed to identify all 34 sessile serrated lesions (100%). They concluded that the setup used in their study has the potential to significantly improve the quality of colonoscopy [56].

Cybernet Corp. developed multiple variations of their AI system called EndoBRAIN. EndoBRAIN-EYE is a CADe system capable of detecting polyps and reducing the change of missed adenomas. Masashi Misawa et al. [57] published their research in Gastrointestinal Endoscopy (2021), where they highlighted the development and validation process of this AI system. For the purpose of training the AI, they acquired a total of 1405 colonoscopy videos from five medical centers. The AI achieved a sensitivity of 90.5% and a specificity of 93.7% for frame-based analysis. They also showed the per-polyp sensitivity for all polyps (98.0%), for diminutive polyps (98.3%), for protruded (98.5%) and for flat polyps (97.0%) [57]. EndoBRAIN-PLUS is a CADx system designed to identify the type of lesion in colonoscopy. In the development and validation study led by Yuichi Mori et al. in 2021 [58], the AI system was tasked to identify three pathological class predictions (cancer, adenoma, or non-neoplastic) for endocytoscopic images obtained at 520-fold magnification. They used 30 cancers, 15 adenomas, and 5 non-neoplastic lesions for the validation test. As a result, the AI system identified the three pathological classes with an overall accuracy of 91.9%. The system managed to differentiate cancer with a sensitivity of 91.8% and a specificity of 97.3% [58]. In collaboration with Olympus Corp., Cybernet Corp. also developed a CADx system dedicated to ulcerative colitis (UC) called EndoBRAIN-UC. This system is based on their previous CADx tool used for differential diagnosis between neoplastic and non-neoplastic polyps [58]. In the study published in Gastrointestinal Endoscopy in 2019, Yasuharu Maeda et al. [59] described the development and validation process of this AI system. The study included a significant number of endocytoscopic images, and the performance of the CADx was determined based on the histological activity of the biopsy samples. The overall diagnostic sensitivity, specificity, and accuracy for predicting persistent histological inflammation were 74%, 97%, and 91%, respectively [59]. The authors considered the 74% sensitivity to be tolerable because, until then, histologic inflammation was difficult to identify with endoscopy [59].

With more and more AI systems approved for clinical use, the opportunity for research and improvement is impressive, with potential future innovations on the way.

## 6. Future Potential of AI in Digestive Endoscopy

Although AI is typically defined as a machine that replicates human behavior and interactions, we are still far from that. A lot of research, time, and funding is invested into developing an AI system capable of doing one basic human function (i.e., identifying a polyp). While the human brain can accumulate a lot of information (i.e., polyp, normal mucosa, normal vascular pattern, normal colon haustra, bowel movement, feces, water, endoscopic instruments, and so on), our brain can also sort that information, eliminating what is non-essential (or what we consider to be “normal”) and focus on the important aspects (i.e., the polyp). At this stage, the current AI systems mentioned in this review are capable of achieving better results than a novice endoscopist but not better than an expert. Hence, the development of a large database and interactive algorithms for AI training may be a potential target for future studies.

Our group, which included colleagues from the Iasi Branch of the Institute of Computer Science of the Romanian Academy, developed a new experimental deep learning system using NVIDIA devices, specifically the Jetson Xavier NX, for real-time object and polyp detection on video colonoscopy and also proposed a method for semantically identifying different areas in colonoscopic frames [60,61,62] (Figure 1 and Figure 2). The human brain can comprehend each individual unit in an image and scrutinize and select it according to a set of criteria while at the same time comprehending how it interacts with the rest of the environment. If this can be achieved by an AI system, the possibilities may be endless.

Although this study offers a significant amount of valuable current information on artificial intelligence in digestive endoscopy, the may limitation is that it is not a systematic review. This means that this study might be prone to subjective selection bias during the literature searches.

## 7. Conclusions

As we approach the end of the first quarter of the 21st century, technology has become an intrinsic part of our daily life. The same applies to the field of medicine, gastroenterology included. AI models have achieved remarkable success in colonoscopy, with the potential to improve digestive disease diagnosis, teaching for novice endoscopists, and even play an important role in large CRC screening programs. In upper digestive endoscopy, AI devices can assist endoscopists as a second pair of eyes, demonstrating an ability to estimate BE with potential neoplastic transformation, identify gastric cancer, and accurately assess gastric blind spots.

Our belief is that AI might soon become an essential tool in the endoscopic lab as the field of digestive endoscopy becomes more and more dependent on newer technology.

## Figures and Tables

**Figure 1 diagnostics-12-00927-f001:**
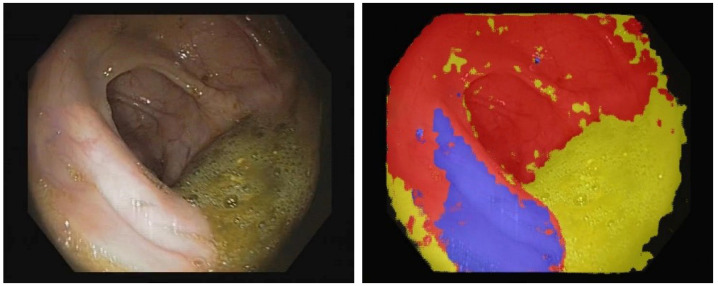
Assigning colors to different components in a colonoscopy frame by deep learning semantic segmentation: mucosa in red, residue in yellow and reflections in blue.

**Figure 2 diagnostics-12-00927-f002:**
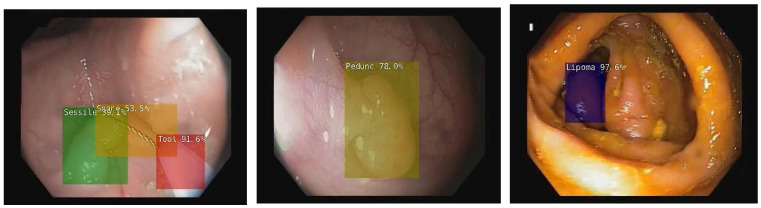
Examples of deep learning colonoscopy detection in real-time on an NVIDIA Jetson Xavier microsystem.

**Table 1 diagnostics-12-00927-t001:** Currently approved colonoscopy computer-assisted tools for commercial use (adapted from Taghiakbari et al., 2021).

Product	Manufacturer	Place of Approval and Year	Computer System Used
EndoBRAIN	Cybernet System Corp./Olympus Corp.	Japan 2018	CADx
EndoBRAIN-EYE	Cybernet System Corp./Olympus Corp.	Japan 2020	CADe
EndoBrain-PLUS	Cybernet System Corp./Olympus Corp.	Japan 2020	CADx
EndoBrain-UC	Cybernet System Corp./Olympus Corp.	Japan 2020	CADx
GI Genius	Medtronic Corp.	Europe 2019	CADe
		United States 2021	
ENDO-AID	Olympus Corp.	Europe 2020	CADe
CAD EYE	Fujifilm Corp.	Europe 2020	CADe/ CADx
		Japan 2020	
DISCOVERY	Pentax Corp.	Europe 2020	CADe
WISE VISION	NEC Corp.	Europe 2021	CADe
		Japan 2021	
CADDIE	Odin Vision	Europe 2021	CADe
ME-APDS	Magentiq Eye	Europe 2021	CADe
EndoAngel	Wuhan EndoAngel Medical Technology Company	China 2020	CADe

CADx—computer-assisted diagnosis. CADe—computer-assisted detection.

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
