# Peer review of "Artificial Intelligence in Digestive Endoscopy—Where Are We and Where Are We Going?"

_diagnostics, 2022, doi:10.3390/diagnostics12040927_

Round 1
Reviewer 1 Report
In this paper, the authors present a review about the artificial intelligence in digestive endoscopy. The detection rate of preneoplastic lesions during colonoscopy may be increased with artificial intelligence assistance. It has also proven useful in detecting signs of ulcerative colitis activity. In upper digestive endoscopy, deep learning models may prove to be useful in the diagnosis and management, like gastroesophageal reflux disease, Barrett’s esophagus and gastric cancer. As it is the case with all new medical devices, there are still challenges in the implementation in daily medical practice.
The topic of this paper is significant and the contents are of reference value for the relative individuals working in this area. The paper expression and organization are acceptable.
Some suggestions:
- In line 62, “have improved…”?
- In line 70, “is considered effective…”?
- In line 227, “To determine if upgrading to an AI system…”?
- In line 251, “Some of the currently issued…”?
- “(narrow binding imaging)” should be explained in the 1st NBI abbreviate.
- What are ASGE, EMA, PMDA, UEG, ESGE? They should be explained.
Author Response
Dear Reviewer,
Regarding your suggestions for our manuscript, we made the fallowing modifications:
- Line 62, “have improved” – we changed it to “has improved” (line 62 in the revised manuscript with simple markup)
- Line 70, “is considered effective” – we changed it to “is considered successful” (line 70 in the revised manuscript with simple markup)
- Line 227, “To determine if upgrading to an AI system” – we changed it to “To decide if purchasing an AI system” (line 233 in the revised manuscript with simple markup)
- Line 251, “Some of the currently issued” – we changed it to “Some of the currently available” (line 267 in the revised manuscript with simple markup)
- Narrow Band Imaging abbreviation (NBI) was explained where it first appeared in the text (lines 101-102 in the revised manuscript with simple markup) as suggested
- The abbreviations ASGE, EMA, PMDA, UEG, ESGE were explained when they first appeared in text as fallow:
- ASGE - American Society of Gastrointestinal Endoscopy (line 126 in the revised manuscript with simple markup)
- EMA - European Medicines Agency (lines 269-270 in the revised manuscript with simple markup)
- PMDA - Pharmaceuticals and Medical Devices Agency (line 270 in the revised manuscript with simple markup)
- UEG - United European Gastroenterology (line 291 in the revised manuscript with simple markup)
- ESGE - European Society of Gastrointestinal Endoscopy (line 83 in the revised manuscript with simple markup)
Beside the reviewers’ suggestions our team also re-analyzed the manuscript and have done mostly minor changes especially regarding some spelling mistakes. We have also changed some words with their synonymous to better emphasize the idea. We changed the size of the table from line 279 and the two images from line 355 to better fit the text on the pages.
One major change was the introduction of a new paragraph that extends from line 253 to 262. We consider the information added here to bring more value to the manuscript and some important information to future digestive AI developers.
All the modification are visible in the revised manuscript with office word track changes set to All Markup.
Thank you for your hard work and dedication.
Sincerely,
Radu Alexandru Vulpoi
Reviewer 2 Report
Dear authors,
I read with extreme interest your article. My major concern is related to the type of article (narrative review) as several similar articles have been published in the last two years. These are my concerns, listed line by line, about the work in its actual form.
-Page 2, line 83 to 89. This concept conflics with latest ESGE guidelines. Diminutive adenomatous polyps should not be left in place. ESGE recommends that all polyps should be resected except for diminutive (≤ 5 mm) rectal and rectosigmoid polyps that are predicted with high confidence to be hyperplastic. Moreover, ESGE recommends retrieval of all resected polyps for histopathological examination. Only in expert centers, where optical diagnosis may be made with a high degree of confidence, a “resect and discard” strategy may be considered for diminutive polyps.
-Page 2, lines 93-94, “This technology is especially useful for beginner colonoscopists, which may achieve results similar to an expert in colonoscopy”. Please include a reference that support this sentence.
-Page 3, line 105, please provide 95% CI linked to OR
-Page 3, line 116 NBI (narrow binding imaging)-CAD: add capital initials
-Page 4, line 161 “these indicators are mostly subjectively assessed” this is not entirely correct. ESGE stated these indicators.
-Study limitations were not mentioned.
Author Response
Dear Reviewer,
Regarding your suggestion on our manuscript we made the fallowing modifications:
- On page 2, lines 83-89 from the unrevised manuscript the paragraph conflicted with the latest ESGE guidelines as the reviewer noted. We changed the entire paragraph in accordance to the latest ESGE guidelines and we also added a new reference ([15]) that supports the new information. The change can be found between lines 83-88 in the revised manuscript with simple markup. Below is the new paragraph:
- "According to current European Society of Gastrointestinal Endoscopy (ESGE) guidelines, all polyps larger than 5 mm need to be removed and sent for histopathology analysis to maximize screening effectiveness. The same thing should be applied to all diminutive polyps (<5 mm) with adenomatous structure. Diminutive polyps located in the rectum and the sigmoid characterized as hyperplastic by a high confidence optical diagnosis tool can be “left in situ” or “resected and discard” [14] [15]. Most digestive endoscopists resect polyps and send them to histopathology because optical diagnosis tools are only available in expert medical institutions. Costs may increase and physicians may become fatigued as a result. Technology development, including AI integration may reduce this setback in the long run."
- Page 2, lines 93-94 the reviewer requested a reference that support the sentence “This technology is especially useful for beginner colonoscopists, which may achieve results similar to experts in colonoscopy”. We provided two references ([17, 18]) and the sentence has move to line 96-98 in the revised manuscript with simple markup. Below are references:
-
[17] Repici, A.; Spadaccini, M.; Antonelli, G.; Correale, L.; Maselli, R.; Galtieri, P.A.; Pellegatta, G.; Capogreco, A.; Milluzzo, S.M.; Lollo, G.; et al. Artificial intelligence and colonoscopy experience: lessons from two randomised trials. Gut 2022, 71, 757–765, doi:10.1136/GUTJNL-2021-324471.
- [18] Xu, Y.; Ding, W.; Wang, Y.; Tan, Y.; Xi, C.; Ye, N.; Wu, D.; Xu, X. Comparison of diagnostic performance between convolutional neural networks and human endoscopists for diagnosis of colorectal polyp: A systematic review and meta-analysis. PLoS One 2021, 16, e0246892, doi:10.1371/JOURNAL.PONE.0246892.
-
- Page 3, line 105: the reviewer requested to provide 95% CI linked to OR. We provided the information as fallowed “OR 1.45 [95% CI 1·14–1·85]; moderate certainty of evidence, and OR 1.54 [95% CI 1·22–1·94]; low certainty of evidence, respectively” which can be found in lines 108-109 in the revised manuscript with simple markup.
- Page 3, line 116, the reviewer asked to add capital initials to “narrow binding imaging” – we added capital initials and corrected the name to “Narrow-Band Imaging” (line 120 in the revised manuscript with simple markup)
- Page 4, line 161, the reviewer noted that the sentence “these indicators are mostly subjectively assessed” is not entirely correct. We change the sentence with “Currently, these indicators are subject to intentional or unintentional manipulation” (lines 166-167 in the revised manuscript with simple markup). Also, in lines 171-172 of the revised manuscript we added the phrase “Using AI together with other computerized management systems”. Thus, in this paragraph we highlighted the potential benefits for data certainty when using computer management systems together with AI.
- We added a paragraph about our study limitation which can be found on page 8, between lines 361-364 in the revised manuscript with simple markup as was suggested by the reviewer. Below is the paragraph:
-
"This study provides a significant amount of current information on artificial intelligence in digestive endoscopy, however, the main limitation is that it is not a systematic review. As a result, the selection of literature for this study may be subject to subjective selection bias."
-
Beside the reviewers’ suggestions our team also re-analyzed the manuscript and have done mostly minor changes especially regarding some spelling mistakes. We have also changed some words with their synonymous to better emphasize the idea. We changed the size of the table from line 279 and the two images from line 355 to better fit the text on the pages.
One major change was the introduction of a new paragraph that extends from line 253 to 262. We consider the information added here to bring more value to the manuscript and some important information to future digestive AI developers.
All the modification are visible in the revised manuscript with office word track changes set to All Markup.
Thank you for your hard work and dedication.
Sincerely,
Radu Alexandru Vulpoi
Round 2
Reviewer 2 Report
Thanks for replying to my comments appropiately.